

# Measurements of Aerosols and Charged Particles On the BEXUS18 Stratospheric Balloon

Erika Brattich[1], Encarnación Serrano Castillo[2], Fabrizio Giulietti[2], Jean-Baptiste Renard[3], Sachi N. Tripathi[4], Kunal Ghosh[4], Gwenael Berthet[3], Damien Vignelles[3], Laura Tositti[5]

[1]Department of Physics and Astronomy DIFA, University of Bologna, 40126 Bologna (BO), Italy
[2]Industrial Engineering Department, University of Bologna, 47121 Forlì (FC), Italy
[3]LP2CE-CNRS/Université d'Orléans, 45071 Orléans, France
[4]Department of Civil Engineering & Department of Earth Sciences, Indian Institute of Technology, 208016, Kanpur, India
[5]Department of Chemistry "G. Ciamician", University of Bologna, 40126 Bologna (BO), Italy

*Correspondence to*: Erika Brattich (erika.brattich@unibo.it)

**Abstract.** This paper describes the aerosol measurements setup and results obtained during the BEXUS18 stratospheric balloon within the "A5-Unibo" (Advanced Atmospheric Aerosol Acquisition and Analysis) experiment performed on October 10[th], 2014 in northern Sweden (Kiruna). The experimental setup was designed and developed by the University of Bologna with the aim of collecting and analyzing vertical profiles of atmospheric ions and particles together with atmospheric parameters

(temperature, relative humidity and pressure) all along the stratospheric ascent of the BEXUS18 stratospheric balloon. Particles size distributions were measured with the MeteoModem Light Optical Aerosol Counter (LOAC) and air ion density was measured with a set of two commercial and portable ion counters. Though the experimental setup was based upon relatively low-cost and light-weight sensors, vertical profiles of all the parameters up to an altitude of about 27 km were successfully collected. The results obtained are useful for elucidating the relationships between aerosols and charged particles between

ground level and the stratosphere with great potential in collecting and adding useful information in this field, also in the stratosphere where such measurements are rare. In particular, the equipment detected coherent vertical profiles for particles and ions, with a particularly strong correlation between negative ions and fine particles, possibly resulting from proposed associations between cosmic rays and ions as previously suggested. In addition, the detection of charged aerosols in the stratosphere is in agreement with the results obtained by a previous flight and with simulations conducted with a stratospheric

ion-aerosol model. However, further measurements under stratospheric balloon flights equipped with a similar setup are needed to reach general conclusions on such important issues.



**Keywords:** 0305 Aerosols and particles; 0394 Instruments and techniques

## 1 Introduction

It is well-recognized that aerosols play a fundamental role in the lower atmosphere as they may affect climate with both a direct effect on absorption and scattering of solar radiation but also an indirect effect through cloud processing (Yu and Turco, 2001; Forster et al., 2007). Aerosols are tightly involved in the atmospheric chemical mass balance, including the stratospheric chemistry through the heterogeneous reactions with nitrogen and halogen species triggering the austral ozone hole through Polar Stratospheric Clouds (PSCs) (Hanson et al., 1994; Deshler, 2008). Aerosol still represents the largest uncertainty in the correct estimate and interpretation of the ongoing change in Earth's energy budget (Boucher et al., 2013, IPCC, 2013; Myhre et al., 2013). In this framework, it is therefore of paramount importance to accurately and systematically collect experimental data such as particle number densities, as well as all the properties shedding light on their nature, their size distribution, and their source in order to define both qualitatively and quantitatively their role, in the troposphere as well as stratosphere.

Stratospheric aerosols are contributed by several sources which determine particle size, composition and morphology, as well as their mean residence time. Historically, the first measurements of stratospheric aerosol were carried out by Junge (Junge, 1961; Junge et al., 1961); stratospheric aerosols drew the attention of scientists during the Cold War owing to the artificial radioactivity released into the stratosphere and returned to the troposphere through the Stratosphere-to-Troposphere exchange processes (Corcho Alvarado et al., 2014; Feely et al., 1966). The monitoring of radioactive fallout from nuclear weapon testing (and in 1964 from the accident of SNAP9A, a nuclear-fueled satellite which released $^{238}$Pu in the upper atmosphere, upon navigation failure, Eisenbud and Gesell, 1997) not only brought about the understanding of the basic dynamic processes coupling the troposphere and stratosphere, but also the discovery of cosmogenic radionuclides which have their maximum production in the stratosphere, mostly in the form of aerosols, still largely employed in the study of vertical exchange between the innermost atmospheric layers (Cristofanelli et al., 2018). Beside radionuclides, the main source of aerosol particles in the stratosphere is through the flux of sulfur bearing molecules into the stratosphere from the troposphere, primarily, OCS (atmospheric carbonyl sulphide), during non-volcanic times, and $SO_2$ from volcanic eruptions: after release, sulfur is oxidized and converted to sulfuric acid which then condenses forming the bulk of the stratospheric aerosol layer (e.g., Kremser et al., 2016). Secondary sources comprehend the outer space, contributing an array of mineral micrometeoritic particles mainly in the solid phase, *in situ* emission from aircrafts (Murphy et al., 1998) and the troposphere itself, through active upward transportation; the troposphere may also act as a source through more localized exchange processes likely mediated by cumulo-nimbus dynamics.

Tropospheric sources include volcanic eruptions, usually occurring on an event basis and with a strong dependency on the event energy (see for example Deshler, 2008 or Murphy et al., 2014), as well as the transport in the Tropical Tropopause Layer (TTL; ~ 12-18 km) of air and (water-insoluble) gases and particles driven by cumulonimbus cloud and further upward transport within the TTL (and in the tropical lower stratosphere) due to the Brewer-Dobson circulation (e.g., Buchart, 2014). Fromm et



al. (2000) also showed that major forest fires can locally inject large number of carbonaceous and potassium rich aerosols in the lower stratosphere, often internally mixed with meteoritic smoke (Hervig et al., 2009; Neely et al., 2011) and solid grains survived from their atmospheric entry (Cziczo et al. 2001; Renard et al., 2005). In this framework, additional evidence of tropospheric contribution has been shown lately by Yu et al. (2017), who indicated a major role of the Asian summer monsoon

in contributing more than 15% of the particles in the stratosphere, potentially emphasizing anthropogenic aerosol contributions. Overall, while recent progress both in stratospheric observations and research suggests the need for increasing attention and consequently for data collection on stratospheric aerosol, most of the investigations on this topic are still mainly focused on volcanic contributions which constitute a sort of baseline to assess behavior and properties of stratospheric aerosols themselves, but also inspire potential as well as very arguable global warming countermeasures under the wide term of geoengineering

(Launder and Thompson, 2009). Indeed, volcanic eruptions strongly influence particle populations in the stratosphere, in particular those from El Chichon in 1982 and from Mt. Pinatubo in 1991 (e.g. Russell et al., 1996), whose monitoring in the course of the past decades brought to assess how stratospheric aerosols may be strongly affected by extreme events until their slow removal, when they reach background concentration levels. Recently, it was shown that even weaker volcanic events, similar to biomass burning, are capable to access at least the lower stratosphere suggesting the need for further investigation

and monitoring (Robock, 2000). As a result, using various measurement techniques such as remote sensing observations from satellites and *in situ* measurements is needed in order to get an updated view of the stratospheric system and its connections with upper and lower layers of the atmosphere (e.g. Steele et al., 1999; Deshler et al., 2003).

It is commonly assumed that stratospheric aerosols are mostly liquid consisting of sulfuric acid from volcanic eruptions, while the upper stratosphere is free of aerosols except for some instances of residual particles from meteoritic disintegration and for

interplanetary grains in very low concentrations (Murphy et al., 2014). Nevertheless, it seems that the stratospheric aerosol content is more complex, both in terms of aerosol concentrations and nature (Renard et al., 2008). Vernier et al. (2011) showed that moderate volcano eruptions can inject a significant number of aerosols into the stratosphere, refilling the aerosol layer episodically. Though less investigated than tropospheric aerosols, extensive details on the properties of stratospheric aerosols are provided in the recent review by Murphy et al. (2014).

In this framework, another physical property still highly underscored, if not in very specialized science fields, is represented by aerosol electrical characteristics. Air conductivity due to the presence of differently sized ions has long been recognized and studied as reviewed in Clement and Harrison (1991), Hirsikko et al. (2011), Harrison and Carslaw (2003). Charged tropospheric aerosols were detected not only in disturbed weather but also in fair weather atmosphere, as resulting from ion diffusion. Charged particles have also been detected in volcanic ashes (Gilbert et al., 1991; Harrison et al., 2010) and in Saharan

dust layers (Nicoll et al., 2011). In the mesosphere, smoke and ice particles are part of the plasma in the D-region and carry positive and negative charges (Rapp, 2009). In the stratosphere, electrified aerosols have been detected in situ for the first time during a balloon-borne aerosol counting measurements (Renard et al., 2013). The observations carried on during that stratospheric flight showed that most of the aerosols are charged in the upper troposphere from altitudes below 10 km and in the stratosphere from altitudes above 20 km, while the aerosols seem to be uncharged between 10 km and 20 km. The





electrification of the aerosols could originate from ion clusters produced mainly in the atmosphere by the interaction of galactic cosmic rays with the atmospheric gases especially in the dense regions of the planetary atmospheres where solar extreme ultra violet radiation is absent (Harrison and Carslaw, 2003).

Major sources of ions in the atmosphere include radon isotopes, cosmic rays, and terrestrial gamma radiation, with a variable
relative contribution depending on the altitude and latitude (Tinsley, 2008): while ionization from turbulent transport of radon and gamma radiation prevail near the Earth's surface and over the continents, ionization due to cosmic rays dominates far away from the continental surface (i.e., over the oceans and in the upper stratosphere/lower troposphere) (Hirsikko et al., 2011) and where also the production of cosmogenic radionuclides is highest (Tositti et al., 2014 and references therein). Both primary and secondary ionization may therefore interact with air components to produce air ions.

While the overall air ion population is largely responsible for the so-called "atmospheric global electric circuit" (Tinsley, 2008), the detection of charged particles in the stratosphere is extremely important since they might have affected both sprite formation and stratospheric photochemistry (e.g., Belikov and Nikolayshvili, 2016 and references therein).

In this framework, the detection of ions and charged particles across the atmospheric column is ever increasingly drawing attention due to the experimental evidence linking ions to nucleation mechanism, firstly proposed by Raes and Janssens (1985).
Requiring a smaller supersaturation of the involved gases, ion-induced nucleation is thermodynamically advantaged over homogeneous nucleation. Even though experiments still do not agree on the relative contributions of ions and neutral nucleation (e.g., Eisele et al., 2006; Suni et al., 2008; Yu, 2010), this effect is at the basis of the proposed link between the flux of ionizing galactic cosmic rays modulated by solar activity and the global cloud cover (Svensmark and Friis-Christensen, 1997; Carslaw, Harrison and Kirkby, 2002) which predicts that an increase in cosmic ray intensity causes an enhancement in
CCN (cloud condensation nuclei) abundance and therefore of cloud reflectivity and lifetime (by suppressing rainfall). This hypothesis stems from the observed enhancement of cloud cover during peaks of high energy radiation leading to enhanced particle formation and growth in the presence of ions: depending on the competition between condensation growth and processes reducing particle concentrations (i.e., coagulation, surface deposition, and in-cloud scavenging), a fraction of those particles may eventually grow into the size of CCN. This mechanism, unlike the aerosol indirect effect, is only driven by
changes in the rates of microphysical processes and acts on a global scale being stronger in regions of low aerosol concentrations.

A second link between galactic cosmic rays and global cloud cover, the ion-aerosol near-cloud mechanism (Tinsley et al., 2000), involves the effects of cloud microphysical properties due to the accumulation of space charge on the tops of clouds. This mechanism is less understood than the former one, but is linked to the hypothesis that variations in cosmic ray ionization
might modulate the fair-weather current generated by the electrical current flowing into clouds leading to a sequence of both micro and macro-physical responses in cloud processing (Dunne et al., 2012).

Observations conducted to study and quantify these effects (e.g., Laakso et al., 2007; Svensmark et al., 2007; Pierce and Adams, 2009) are often incomplete and non-conclusive leaving model simulations as the most convenient source of information. The search for a link between CRs (Cosmic Rays) and cloud formation is also one of the main drivers for the



CLOUD experiment being conducted at CERN since 2009 where a chamber filled with atmospheric gases is crossed by charged pions that simulate ionizing CRs. While some preliminary results suggested that indeed IIN (Ion Induced Nucleation) is a relevant factor to determine nucleation rates in the upper troposphere (e.g., Kirkby et al., 2011), recent results show that cosmic ray intensity cannot meaningfully affect climate via nucleation (Dunne et al., 2016) while others indicate that IIN of

pure organic particles constitutes a potentially widespread source of aerosol particles in terrestrial environments with low sulfuric acid pollution (Kirkby et al., 2016).

Stratospheric balloon research devoted to the collection of aerosol profiles vs. height have traditionally been carried out in the last decades with the aim of elucidating properties and processes of this fundamental air component. While as a rule, in situ observations, either onboard stratospheric balloons either onboard aircrafts, are fairly demanding and costly (e.g., Hunton et

al., 2005; Curtius et al., 2005; Andersson et al., 2013; Watanabe et al., 2004; Murphy et al, 2014; Sugita et al., 1999; ; Matsumura et al., 2001; Hervig and Deshler, 2002; Kasai et al., 2003; Deshler et al., 2003; Shiraishi et al., 2011), the present paper aims at promoting the formation of young researchers in the spirit of the BEXUS initiative (Balloon-borne Experiments for University Students, see below), but also in elaborating effective and "relatively" cheap experiments to fulfill the need of experimental vertical profiles of aerosol data useful for filling knowledge gaps in atmospheric and climatological research.

The "A5-Unibo" (Advanced Atmospheric Aerosol Acquisition and Analysis) experiment designed by the University of Bologna has been developed with the purpose of collecting and studying vertical profiles of atmospheric ions and particles in addition to atmospheric parameters (temperature, relative humidity and pressure) all along the flight path of the BEXUS18 stratospheric balloon, using a relatively low-cost and low-weight setup compared to the conventional instrumentation onboard stratospheric balloons. This paper describes the setup of the experiment and the measurements obtained during the flight.

## 2 Instrumentation

The "A5-UNIBO" experiment was flown from SSC, Esrange Space Center in northern Sweden (Kiruna; 67°53'N, 21°04'E) on October 10th, 2014 with the BEXUS18 stratospheric balloon under the REXUS/BEXUS program. The program was realized under a bilateral Agency Agreement between the German Aerospace Centre (DLR) and the Swedish National Space Board

(SNSB). The Swedish share of the payload is available through collaboration with the European Space Agency (ESA).

The balloon was a Zodiac BL-DD-12SF-404-ZIT filled with helium gas with a volume of 12,000 m$^3$ and a diameter of 14 m. The flight lasted for 3 hours from 08:48 to 12:00, and reached a maximum altitude of 27.2 km with an average ascending speed during climbing of 3.5m s$^{-1}$. Due to onboard electric problems, no data was available between the altitude range of 18.5 km and 20 km. The floating time in the stratosphere was 1 hour and 8 minutes. The balloon eventually landed in Finland where it

was promptly retrieved and safely brought back to the Esrange Space Center facilities the day after the flight.

Apart from the "A5-UNIBO" experiment whose results and setup are herein described, the whole payload of the stratospheric balloon was quite big and included a wide range of different experiments: the AFIS-P (Antiproton Flux In Space-Prototype),



ARCA (Advanced Receiver Concepts for ADS-B), COUGAR (Control of Unmanned Ground Vehicle from Higher Altitude in near Real Time), and POLARIS (POLymer-Actuated Radiator with Independent Surfaces)

## 2.1 Aerosol measurements

Particles size distribution vertical profile was measured by the Light Optical Aerosol Counter (LOAC) (MeteoModem Inc.),

an optical particle counter/sizer (Renard et al., 2016a,b based on scattering measurements at angles of 12° and 60°. The instrument is light (250 g total weight including the pump) and compact enough to perform measurements on board of all kinds of balloons. As described in detail in Renard et al. (2016a,b), the combination of the measurements at two scattering angles provides both the determination of the particle size distribution and an estimation of the typology of particles in 19 size classes from 0.2 to 100 µm: briefly, while the measurement at 12° scattering angle does not depend on the refractive index of the

particles and enables for accurate size determination and counting, the measurement at 60° scattering angle is strongly sensitive to the refractive index of the particles, giving information on the nature of the particles. LOAC has already performed more than 150 flights in the stratosphere since 2013.

## 2.2 Ion measurements

Air ion densities vertical profile was measured by means of two air ion counters (ALPHALAB Inc.), respectively for positive

and negative ions. These instruments are handheld meters designed to measure ion density, i.e., the number of ions per cubic centimetre (ions/cc) in air. The instrument is a ion density meter, based on a Gerdien Tube condenser design, and containing a fan which draws air through the meter at a calibrated rate and is able to count the number of positive/negative ions when the voltage applied to the outer cylinder is positive/negative respectively. The air ion counter can be used for the detection of natural and artificial ions. Natural ions include those generated from the decay of radioactive minerals and radon gas, fires,

lightning, and evaporating water, and finally ions associated with storm activity. These devices are light and small enough (305 g each; 160 x 100 x 55 mm) to be mounted on balloons; their measurement resolution extends up to 200000 ions cm$^{-3}$. These probes are equipped with a fan to create an air flow of 24 l min$^{-1}$ throughout the gondola. Air ambient ions are diverted from the flow and collected on a plate which returns a voltage output proportional to the number of ions. The air is then expelled downwards through the bottom plate.

The use of these small commercial probes based on Gerdien tube meter (see for example Aplin and Harrison, 1999) for stratospheric balloon experiments is unprecedented, therefore quality control procedures for ion quantification are not yet available; however, the instrumental performance of the air ion counters at stratospheric conditions was tested in pre-flight lab experiments simulating stratospheric conditions.

The ion counters were operated for offset values first in a vacuum changing the pressure between 1000 and 5 hPa and

subsequently in a thermal chamber between 15°C and -60°C.

The result of these tests was that both of the air ion counters worked properly in low-pressure environments, while the offset was found to be independent of external pressure. It is also worth pointing out that during this experiment the fan flow rate



was expected to remain constant during the balloon's ascent phase. Even though the flow rate actually could monotonously decrease with increasing altitude, thus leading to underestimations in in ion density concentrations with increasing altitude, preliminary tests performed to assess the flow rate dependence on pressure were not conclusive and did not provide a satisfactory working curve. However, even if ion concentration variations might be biased by the pressure dependence of the

air flow rate, this bias does not affect the relative variations of concentrations (local strong increases or decreases), which in fact variations seem to be consistent, as shown and discussed later on.

### 2.3 Temperature, relative humidity measurements and other instrumentation

The BEXUS-18 gondola was also equipped with a Parallax MS5607 altimeter module for pressure readings, which was successfully tested at 120,000 feet. A humidity sensor HIH9120-021 was used to record the vertical profile of relative humidity.

A temperature sensor LM35DZ was mounted on the electronic board to control the internal temperature. External temperature and GPS data was instead retrieved by the Esrange Balloon Service System (EBASS), a Telemetry/Telecommand (TM/TC) service system for stratospheric balloons developed by SSC and DLR in 1998.

All the sensors and the instruments onboard (especially the air ion counters and the LOAC) were tested prior to the stratospheric flight in a vacuum chamber to ensure their proper functioning at ambient pressures from 1000 to 5 mbar. In addition, after

complete assembling of all the probes, the whole experiment was put into a thermal chamber to ensure its proper working at low temperatures conditions. All the tests confirmed the correct performance of the experiment under stratospheric conditions. For a detailed test related to the LOAC performance, the reader is referred to Renard et al. (2016a).

An Arduino MEGA 2560 microcontroller was used for data acquisition from sensors and instruments. The sensors were connected to ARDUINO through a hardware interface and two stacked boards. An Arduino Ethernet Shield was used to

connect the Arduino MEGA 2560 board to the BEXUS telemetry system. Two additional electronic boards were designed for the respective control of the heating system which kept the temperature of the key components above 0°C and of a power control unit which fed the probes and sensors with the required voltage and current.

Data acquired by the on-board unit, including ambient data and internal sensors, was collected with a 10 second time resolution; the data was transmitted to the Ground Station and displayed via HID through a graphical interface. The data was then

integrated over 60 seconds and only data acquired during the ascent and floating phases was analyzed.

### 3 Numerical simulations

Model calculations have been used to quantify the electrification of aerosols with a stratospheric ion–aerosol model in the altitude range of LOAC measurements. Ion clusters in the atmosphere are produced primarily by interaction of galactic cosmic-rays with atmospheric gases, especially in the dense regions of planetary atmospheres where extreme solar ultraviolet radiation

is absent (Harrison and Carslaw, 2003). A high fraction of the cosmic ray (1 GeV) energy flux is typically carried by particles of high-kinetic energy. The peak ion production rate by this process has been found to be generally located at altitudes between





14 and 17 km (Rawal et al., 2013), which is our major study area, and the ion pair production rate is calculated using the statistical model of O'Brien (2005), considering $SO_4^{2-}$ and $NH_4^+$ as the most abundant ion clusters produced by this process (Renard et al., 2013). Other sources radon isotopes and terrestrial gamma radiation) can be included for the further improvement of the model simulation as one of the future scopes of the study.

This ion pair production rate is calculated using the statistical model developed by O'Brien (2005) with the major ions considered being $SO_2^{-4}$ and $NH^{+4}$. Electrons are not included in the model as they recombine with positive ions and uncharged molecules very rapidly, and are consequently not available to interact with aerosols. The charging of aerosols is calculated using charge balance equations as described by Michael et al. (2008, 2009) and Tripathi et al. (2008). The charged particles are constantly interacting with each other resulting in changes in initial charge and size distribution with time. As this is a

bipolar interaction, it is expected that charge distribution will be wider than initial distribution with time (Ghosh et al., 2017).

$$\frac{dn^+}{dt} = q - \alpha n^+ n^- - [n^+ \sum_{j=rmin}^{rmax} \sum_{i=-p}^{p} \beta_{i,j}^+ N_{i,j}] \qquad (1)$$

$$\frac{dn^-}{dt} = q - \alpha n^+ n^- - [n^- \sum_{j=rmin}^{rmax} \sum_{i=-p}^{p} \beta_{i,j}^- N_{i,j}] \qquad (2)$$

In equation (1) and (2), $n^+$ and $n^-$ represent positive and negative ion concentrations, respectively, $q$ is the ion pair production

rate, $\beta$ is the ion–ion recombination coefficient, $N$ is the aerosol concentration, and $\alpha$ is the ion–aerosol attachment rate. Here the radii of the aerosols vary from size *rmin* to *rmax*, and the maximum number of elementary charges an aerosol can own is $p$. The aerosol concentration for any size and charge is calculated by Eq. (3), where $i$ represents the number of elementary charges on a particle for $j$, the associated radius bin.

$$\frac{dN_{i,j}}{dt} = \beta_{i-1,j}^+ N_{i-1,j} + \beta_{i+1,j}^- N_{i+1,j} n^- - \beta_{i,j}^+ N_{i,j} n^+ - \beta_{i,j}^- N_{i,j} n^- + \frac{1}{2} \sum_{i=-p}^{p} \int_0^v K_{j-v,v}^{i-1,1} N_{j-v}^{i-1} N_v^1 dv -$$

$$N_{i,j} \sum_{i=-p}^{p} \int_0^v K_{j,v}^{i,1} N_j^i N_v^1 dv \qquad (3)$$

In equation (3), the first two terms on the right-hand side are the probability of interaction between the ions and aerosols of any particular charge and size, and the last term is for growth of that particle due to the charge-particle coagulation process. $K$ is the charge particle coagulation coefficient (probability of collision between two charged particles). $v$ is the aerosol particle

volume assuming all particles have a spherical shape, $N_s$ is the number of aerosol particles for any particular size. Full details are provided in Ghosh et al. (2017).



The model is run for amount of charges of any particular size of particle ($q$) running from +20 to -20. The ion–aerosol attachment coefficients (β) are calculated in different ways depending on the relative size of the particles with the ionic mean free path. The calculation depends in particular from the different regimes, i.e. diffusion, free molecular and transition. Hoppel and Frick (1986) developed a method to calculate in all three different regimes. The major requirements for this calculation

are the ionic mobility and mean free path, which are calculated using the expressions given by Borucki et al. (1982). The charge coagulation coefficient $K$ was calculated from diffusional force (including vertical diffusion), turbulent shear force, turbulent inertial force along with electrostatic force due to charge on particles (Ghosh et al., 2017).

A polydispersed distribution of aerosols is used in the model and is obtained from the LOAC observation. The LOAC measured data was used for calculation of different input parameters, like ionic mobility, charge particle mobility, charge particle

coagulation, ion-aerosol attachment coefficient and ion-ion recombination, which are the global input parameters for the overall model (Global Electrical Circuit Model (GEC), as described in Rawal et al. (2013). The model also uses temperature and pressure measured during the experiments. We added charged particle coagulation model to the Renard et al. (2013) model, as it is close to accurate simulation scenario. The charge balance equations are solved by implicit numerical method to obtain concentrations of positive ions, negative ions, uncharged aerosols and charged aerosols, for the steady state.

**4 Results**

Fig. 1 shows the comparison between the vertical profiles of relative humidity and temperature measured during the flight with those measured with the radiosonde sounding at Kandalashka (67.15 N, 32.35 E, 25m asl; Russia), 12UTC. The comparison of the temperature profiles shows good agreement in the troposphere with a small inversion layer close to the ground and the starting of the inversion typical of the tropopause located at about 11 km up; The comparison of the temperature profile in the

stratosphere presents instead major differences: in fact, while onboard the Bexus flight the temperature remains almost constant in the stratosphere up to an altitude of 26 km and presents a sudden and strong increase around 26-27 km, the temperature profile measured at Kandalashka presents a small decrease until about 25-26km, typical of profiles of this time of the year in the Arctic region. The reason of such discrepancy in the stratosphere might be due to solar heating and perhaps to heat/solar light reflection from other instruments/structures of the gondola. The comparison of the relative humidity profiles present

instead major differences already in the troposphere: in particular, the strong dryness in the Planetary Boundary Layer (PBL) detected onboard the Bexus flight (less than 20%) with a further decrease until the altitude of about 5 km is probably due to the slow response of the relative humidity sensors used onboard. Since standard radiosonde measurements of relative humidity are only reliable in the troposphere above temperatures near -40°C, whereas below these temperatures and in the stratosphere special instrumentation for stratospheric water vapor measurements is needed (Berthet et al., 2013; Tomikawa et al., 2015),



measurements of relative humidity above the tropopause are reported only for the Kandalaksha profile, which have to be treated with care nevertheless.

Fig. 2 reports the vertical profiles for the cumulative aerosol particle number density obtained by summing up the data from all the size bins collected by the LOAC (in black) together with the negative (blue) and positive (red) ions during the ascent.

A sliding smoothing (i.e., each point is simply replaced with the average of m adjacent points) is applied to suppress small scale fluctuations.

Fig. 3 reports the vertical profiles of aerosol size distribution for each size bin measured by the LOAC instrument, which shows that all fine particles (< 1µm) presented the same vertical variation, while larger particles, besides presenting lower number concentrations as expected, presented a different vertical profile, with the presence of an abrupt increase in the PBL and then

at 10 km less evident in finer particles.

In Fig. 4 we report the average variation of particle size distribution with altitude. The five size distributions depicted have been determined empirically by averaging the LOAC data over five temperature intervals along the height profile (see Fig. 1) according to roughly coherent atmospheric layers, as obtained during the BEXUS-18 experiment. The intervals chosen were respectively: 0 - 404 m; 650 - 1519 m; 1765 – 10118 m; 10650 - 25044 m; and 25289 – 27191m.

In practice, they correspond respectively to lower, upper PBL, free troposphere, tropopause/lower stratosphere, and mid stratosphere, which as known is characterized by a marked increase in temperature owing to the ozone absorption of longer wavelength UV radiation. However, the marked temperature increase recorded at float in our measurements is probably due to instrumental errors since the relative speed between the balloon and the air is close to 0 there (no ventilation). While there is a steady decrease of particles in all the size bins as the altitude increases, in the intermediate tropospheric groupings an

increase in the coarse particle bins around and above 10 µm is observed.

## 5 Discussion

The data collected shows that there is a steady decrease in the particle number density with height for all the size bins determined. At all heights sub-micron particles are the most numerous, though coarse particles show a relative, sensitive increase in the upper PBL and free troposphere. At an altitude of 1 km, LOAC typology measurements indicate the presence

of a thin layer, less than 100m width, of transparent particles, possibly droplets. At the other altitudes the typology measurements indicate optically absorbing and semi-absorbing particles, probably related to the presence of minerals (dust). In particular, above the tropopause, almost all of the particles detected by LOAC were smaller than 1 µm, and their typology measurements indicate both the occurrence of stratospheric liquid droplets and the presence of optically absorbing material (i.e., internally or externally mixed particles): even though LOAC typology measurements cannot provide precise information

on particles' chemical composition, is in agreement with results from aircraft observations in the lower stratosphere (e.g. Schwartz et al., 2006, Murphy et al., 2014), which showed that while sulfate particles dominate the aerosol composition in the stratosphere, other sources producing absorbing particles also contribute. The vertical profiles of integrated concentration of





aerosols > 200 nm and of ions (Figure 2) present interesting features. Firstly, positive charges are only present relatively close to the ground, which is in agreement with previous observations (e.g., Li et al., 2015), even though we cannot exclude that the complete absence of positive charges at upper levels derives from a failure of the positive ions counter during the flight. Even though we are performing more flights with a similar instrumental setup in order to compare and provide evidence of our findings, this general behavior is also consistent with previous observations, showing that ionization from turbulent transport of radon (positively charged product ions) and gamma radiation (negative ions) prevail close to the Earth's surface, whereas ionization from cosmic rays (negative ions) dominates away from the continental surface (upper troposphere and above) (Hirsikko et al., 2011), both ion distribution playing a basic role in the terrestrial global circuit (Tinsley and Zhou, 2006). Indeed, preliminary results of a stratospheric flight with the ion counters performed on 8 April 2017 in Australia show and confirm the detection of only positive ions in the lower troposphere, while both polarities, with a prevalence of negative ions, were present at upper levels. Secondly, vertical profiles of particles and ions present the same general structure above the tropopause, including an enhancement in the 20-25 km altitude range. None of these profiles are similar to the temperature, humidity, and pressure profiles, which can exclude the possibility of instrumental contamination by these atmospheric parameters. Moreover, the ion concentration variations between 10 and 20 km cannot be linked to the decrease in the airflow fan, whose precise dependence on pressure cannot be correctly estimated for the time being as previously pointed out in the material and methods section. Even though the absolute values could be biased from this effect, the relative variations seem to be real. In particular, Spearman's correlation coefficient (a nonparametric measure of rank correlation, where nonparametric means not based on parameterized families of probability distributions) (Table 1) indicates a strong negative correlation (i.e., anticorrelation) of negative ions with fine particles, a behavior which might have resulted from ion induced nucleation and in particular from the proposed association between cosmic rays and ions ("ion-aerosol clear-air" mechanism), although we are aware that nucleation concerns particles in the 1-2 nm range, and further growth is governed by condensation. Our results are also in agreement with previous observations showing that, in general, negative ions more efficiently promote nucleation than positive ions (Eisele et al., 2006; Suni et al., 2008; Svensmark and Friis-Christensen, 1997).

The weaker, but positive correlation between positive ions and coarse particles might instead arise from their simultaneous detection closer to the Earth's surface.

The number of (negative) charges as well as of particles strongly increases above 20 km. The maximum value around 20 km corresponds to the region of maximum ionization (Regener-Pfotzer maximum (Regener and Pfotzer, 1935)) and was previously observed by Harrison et al. (2014) observing count rates through Geiger counters on standard meteorological balloons. Stratospheric ion-aerosol model simulations can be used to quantify and explain the electrification of the aerosols (Rawal et al., 2013). The simulated profile shows that more than 75% of aerosols are charged above the altitude of 5 km (Figure 5). This result is in agreement with the presence of the charged liquid and/or solid particles detected by ion detectors and could be used as an estimate of the vertical variability of their percentage. The measurements presented here are also in general good agreement with the unique previous direct detection of charged stratospheric aerosols of Renard et al. (2013). In addition, they



reveal a "depletion layer" of poorly charged aerosols from the tropopause to an altitude of about 20 km, where the charged fraction drops at about 1%, similar to the one previously detected by Renard et al. (2013).

In particular, as from Figure 5 b, it is clear that as from model simulations, fine particles are the ones contributing to the largest variations in the fraction of the charged fraction, while coarse particles, when present, are mostly charged, confirming the

calculations made by Renard et al. (2013) (see their figure 4).

# 6 Conclusions

The A5-UNIBO experiment flown under a stratospheric balloon seems to have confirmed the previous detection of charged aerosols in the stratosphere and a possible vertical variability. In particular, the results show coherent vertical profiles for particles and ions, with a particularly strong correlation between negative ions and fine particles, possibly resulting from

proposed associations between cosmic rays and ions as previously suggested. Due to the important implications of charged aerosols on the high-energy phenomena (sprites, blues jet and elves) in the middle stratosphere (Fullekrug et al., 2016) and of ions in nucleation mechanisms, further stratospheric balloon-borne measurements of charged particles are necessary. Poly-instrumented gondolas with aerosols counters for the estimate of the percentage of charged particles, positive and negative ions counters, and Geiger counter, will help to better evaluate the direct link between cosmic rays, ions, and the charged

aerosols. Furthermore, the addition of small condensation particle counters able to characterize particles in the 1-2 nm range could help to gain precise information on nucleation, which here was only derived and could not directly observed. In particular, since both the present flight and the Renard et al. (2013) flight were performed for altitudes below 27 km, an experimental campaign comprehensive of many flights at higher altitude, up to the maximum altitude reachable with stratospheric balloons (around 40 km), is necessary to better document the vertical evolution of the charged aerosols. Also, the

analysis of the results obtained during such future flights will be helpful in answering open questions raised in this and previous flights:

- Are the stratospheric aerosols always charged?

- Is there variability in the percentage of charged aerosols, for example with latitude, season, or solar eruptions?

- Is the percentage of charged aerosols dependent on the nature of the aerosols (liquid droplet, ash from volcanic

eruptions, meteoritic mineral material, or carbonaceous particles from Earth and space)?

- Is the "depletion layer" of poorly charged aerosols above the tropopause, not expected from modeling, a transient phenomenon or a permanent feature?

Such aerosols measurements could have implication on climate and atmospheric chemistry issues, but also on the atmospheric electricity and high energy phenomena such as sprites, blue jets and elves that are not yet well understood.

**Acknowledgements**

This work was designed and developed within the collaboration of the Flight Mechanics Laboratory (Prof. Fabrizio Giulietti)
and the Environmental Chemistry and Radioactivity Laboratory of the Department of Chemistry "G. Ciamician" (Prof. Laura
Tositti) of the University of Bologna as main supporters. In particular, the experiment was flown onboard the BEXUS18
stratospheric balloon under the REXUS/BEXUS program, supported under a bilateral Agency Agreement between the German Aerospace
Centre (DLR) and the Swedish National Space Board (SNSB). The Swedish share of the payload is available through collaboration with the
European Space Agency (ESA). Every team member of the A5-UNIBO experiment is acknowledged for his/her essential role for
the success of the experiment: Encarnaciòn Serrano Castillo (Team Leader and System Engineer), Riccardo Lasagni Manghi
(Verification and Testing engineer), Erika Brattich (Data Analysis, Scientific expert), Igor Gai (Ground Station engineer), Danilo
Boccadamo (Power engineer), Paolo Lombardi (Mechanics), Alice Zaccone (Software engineer), Abramo Ditaranto (Electronics
engineer), Luca Mella (Software engineer), and Marco Didonè (Thermal engineer). We also acknowledge: a) institutional supporters:
DLR, Rymdstyrelsen, SSC, ESA Education Office, EuroLaunch, ZARM; b) private companies and associations: AlphaLab Inc.,
Boxer, Iacobucci HF Aerospace, Icos, CNA Forlì-Cesena, Dogcam, Gruppo SDS, Società Italiana di Medicina Generale, Plastica
Panaro, Bustaplast, Bellini Tiziana, Mascherpa. We also thank Sergio Brattich for a thorough revision of the English language of the
manuscript. Erika Brattich also thanks the Department of Biological, Geological and Earth Sciences of the University of Bologna
for grant support during her PhD study, during which the experiment was developed, and the Department of Chemistry "G.
Ciamician" of the University of Bologna, for support during her post-doc.

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




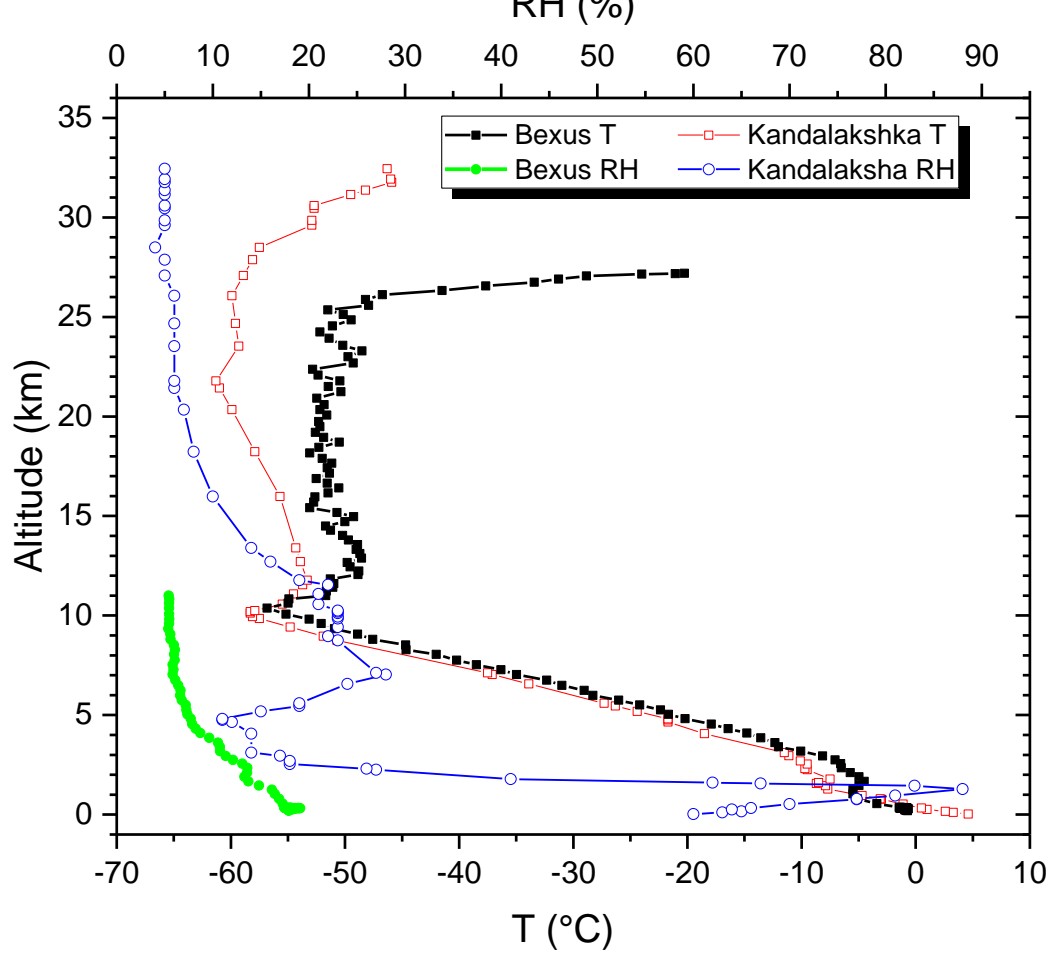

**Figure 1: Vertical profiles of ambient external temperature (Temp) and relative humidity (RH) as measured along the BEXUS18 stratospheric flight on 10 October 2014 and by the radiosounding from the Kandalashka station (67.15N, 32.35E) on 11 October 2014 at 00UTC.**



**Figure 2: Vertical profiles of integrated aerosols concentration, for aerosols greater than 200 nm (Black line), and of positive (red line) and negative ions (blue line).**



**Figure 3: Vertical profiles of particles size distributions measured by the LOAC particle counter as part of the A5-Unibo experiment on the BEXUS18 stratospheric flight.**



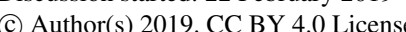


**Figure 4: Average size distribution of aerosol particles as a function of height during BEXUS18 stratospheric flight. The five curves are obtained averaging the aerosol number densities as a function of the atmospheric layers pointed out by the temperature profile as follows: 1: 0-404 m (black line); 2: 650-1519 m (red line); 3: 1765-10118 m (blue line); 4: 10650-25044 m (pink line); 5: 25289-27191 m (green line).**





**Figure 5: Vertical profile of the simulated fraction of charged particles: a) total; b) in the different size classes.**



| | Positive Ions | Negative ions |
|---|:---:|:---:|
| **Positive Ions** | 1.00 | |
| **Negative ions** | -0.28 | 1.00 |
| **0.2-0.3** | *0.44\** | **-0.67\*** |
| **0.3-0.4** | *0.45\** | **-0.75\*** |
| **0.4-0.5** | *0.40\** | **-0.71\*** |
| **0.5-0.6** | 0.33\* | **-0.80\*** |
| **0.6-0.7** | 0.34\* | **-0.62\*** |
| **0.7-0.9** | 0.28 | **-0.76\*** |
| **0.9-1.1** | 0.01 | **-0.62\*** |
| **1.1-3.0** | 0.31\* | -0.22 |
| **3.0-5.0** | 0.15 | -0.07 |
| **5.0-7.5** | 0.12 | -0.28\* |
| **7.5-10.0** | 0.33\* | 0.23 |
| **10.0-12.5** | 0.17 | -0.10 |
| **12.5-15.** | 0.36\* | |
| **15.0-17.5** | 0.36\* | 0.27\* |
| **17.5-20.0** | 0.36\* | |
| **20.0-22.0** | 0.36\* | |
| **22.0-30.0** | 0.32\* | -0.18 |
| **30.0-40.0** | 0.24 | |
| **40.0-50.0** | | 0.18 |
| **Aerosols > 200 nm** | *0.45\** | *-0.59\** |

**Table 1: Spearman's correlation coefficient between ions and number of particles in the different size classes. The asterisk \* indicates significant (0.05 significance level, i.e., p<0.05) correlation coefficients; bold indicates strong (R > 0.6 in absolute value) correlation, while italics indicates weaker (0.4 < R < 0.6 in absolute values)..**

