# Peer review of "Measurements of Aerosols and Charged Particles On the BEXUS18 Stratospheric Balloon"

_Annales Geophysicae, 2019_

## Referee Comment (RC1) · Anonymous Referee #1 · 7 Mar 2019

The contribution by Erika Brattich and colleagues reports the measurement and modelling of charged aerosols in the stratosphere. The manuscript is very well written, logically constructed, easy to follow and informative. The manuscript starts with a substantial review of the relevant literature, followed by a thorough theory section that explains the basis for the simulations. Compared to these first two sections, the consecutive section on the experimental results is rather terse and provides little guidance to the reader as to how the individual findings reported in the list of Figures contribute to the key points of the paper. As a result, it would be helpful to expand this section to make the narrative more clear. It is also somewhat surprising that Fig. 5 is not listed in this section, perhaps because it is not considered to be a result of the conducted work. The final sections with the discussion and conclusions emphasise to a large degree the agreement of the findings with previous work. While it is undoubtedly important to put the findings of this study into context, it makes is harder for the reader to appreciate the novelty of the presented work which becomes less clear. It therefore appears to be beneficial for these two sections to distinguish more clearly between known facts and novel findings. Besides this apparent imbalance between the first and second part of the manuscript, I think it is a valuable contribution to the scientific literature as the current knowledge on charged aerosols in the stratosphere and their spatiotemporal variabilities is somewhat limited at present. Some minor suggestions on how to improve the manuscript are given below.

(1) Fig 2: The concentrations of negative ions appear to be large compared to previous findings. Is there any explanation for this? It is also not clearly explained how the total concentration of aerosols can be smaller than the concentration of negative ions. Is the reader supposed to infer from this that the aerosols <200 nm mainly contribute to the negative ions?
(2) Fig 3: 19 channels are listed in the legend, but only 8 height dependent traces can be distinguished. It is practically impossible to infer any useful information for the PBL.
(3) Fig 4: The x-axis labels are rather sparse and could be more populated.
(4) Fig 5: Again, only 9 curves are shown for 19 channels listed in the legend, as in Fig. 3. Would it not be better to combine some of these channels for the benefit of clarity?
(5) The arrangement in the table appears somewhat unfortunate to me. The first two rows seem to be unrelated to the remainder of the table and the table deserves a heading to state the unit (nm) for the first column and a symbol with unit for the second column.
(6) The acknowledgments have distinct font variations disturbing this reader.

---

## Referee Comment (RC2) · Anonymous Referee #2 · 4 Apr 2019

General comments:

This study reports the aerosol measurements obtained by the BEXUS18 stratospheric balloon flight. Its distinct feature is being equipped with ion counters. A role of ion chemistry in the stratosphere is still an open question and could be essential for the understanding of the atmospheric impact from the space. This topic is suitable for ANGEO. However, information on the instruments of aerosol and ion measurements is completely lacking in this manuscript, so that it is impossible to evaluate whether their observations are reliable or not. Thus I recommend a rejection of this manuscript. Detailed comments are given below.

Specific comments:

[Figure]

-Temperature and pressure measurements

The authors showed results of temperature and pressure measurement in Fig. 1, but it is found that their results are not so reliable compared to nearby radiosonde observation. Including a radiosonde in their payload does not look difficult, so that I am wondering why they did not do it. In addition, they used temperature and pressure data in their model calculation. Is it really meaningful to use such unreliable data? They need to show how sensitive their model calculation is to temperature and pressure errors.

-Aerosol measurements

Their aerosol instrument, LOAC, has been used in 150 flights, so that its precision, resolution, etc. should be well known. However, they did not give those information at all in the manuscript. Although they mentioned an existence of the thin aerosol layer with a thickness of less than 100m at p.10, l.25, I cannot judge whether this instrument has a vertical resolution high enough to detect such a layer.

-Ion measurements

They mentioned that the performance of their ion measurement was checked by pre-flight lab experiments, but it is not shown in the manuscript at all. Thus I cannot judge whether their ion measurements are reliable or not.

-Average

In order to show the aerosol data, they often used arithmetic mean/smoothing. Since aerosol density changes by several orders of magnitude, the arithmetic mean strongly depends on the largest value. Geometrical mean or median filter would be better to represent aerosol distributions.

-Figs. 3 and 4

What is dN/dlog(D) in Fig. 3? A caption of Fig. 4 does not correspond to Fig. 4 about

their y-axis.

-Eq. (3)

Units look different between the terms. Probably some variables are missing.

---

## Author Comment (AC1) · 16 Apr 2019

Dear Editor,

Please extend our deepest acknowledgements to Anonynous Reviewer #1 for his/her comments on our work.

Please find below in italic the comments received by Anonymous Referee #1 after submission to ANGEOD, followed by our replies in normal style.

**Anonymous Referee #1**

*The contribution by Erika Brattich and colleagues reports the measurement and modelling of charged aerosols in the stratosphere. The manuscript is very well written, logically constructed, easy to follow and informative. The manuscript starts with a substantial review of the relevant literature, followed by a thorough theory section that explains the basis for the simulations. Compared to these first two sections, the consecutive section on the experimental results is rather terse and provides little guidance to the reader as to how the individual findings reported in the list of Figures contribute to the key points of the paper. As a result, it would be helpful to expand this section to make the narrative more clear. It is also somewhat surprising that Fig. 5 is not listed in this section, perhaps because it is not considered to be a result of the conducted work. The final sections with the discussion and conclusions emphasise to a large degree the agreement of the findings with previous work. While it is undoubtedly important to put the findings of this study into context, it makes is harder for the reader to appreciate the novelty of the presented work which becomes less clear. It therefore appears to be beneficial for these two sections to distinguish more clearly between known facts and novel findings. Besides this apparent imbalance between the first and second part of the manuscript, I think it is a valuable contribution to the scientific literature as the current knowledge on charged aerosols in the stratosphere and their spatiotemporal variabilities is somewhat limited at present.*

We thank the reviewer for his/her constructive comments. We have included Fig.5 in the narrative of the experimental results section. In addition, the final sections (Sections 4 Results, 5 Discussion, and 6 Conclusions) were revised in accordance to the guidelines provided by the reviewer

*Some minor suggestions on how to improve the manuscript are given below.*

*(1)Fig 2: The concentrations of negative ions appear to be large compared to previous findings. Is there any explanation for this? It is also not clearly explained how the total concentration of aerosols can be smaller than the concentration of negative ions. Is the reader supposed to infer from this that the aerosols <200 nm mainly contribute to the negative ions?*

The detection of high concentrations of negative ions is probably due to the fact we added a separate suitable instrument for measuring ions in this flight. Because of the LOAC lower limit of detection of aerosols at 200 nm, we can expect to have ions concentrations greater than the aerosol detected total concentrations due to the presence of aerosols with aerodynamic diameter less than 200 nm. Since the number of ions is greater than the detected aerosol, this indicates that aerosol smaller than 200 nm are the main contributor to negative ions. This comment was added in the revised version of the manuscript.

*(2)Fig 3: 19 channels are listed in the legend, but only 8 height dependent traces can be distinguished. It is practically impossible to infer any useful information for the PBL.*

The text in the revised version of the manuscript was changed to better describe the vertical profiles of the different particles' sizes presented in the Figure. Information on the fact that the information on the PBL, partially commented but out of the scope of the paper, was also added. The combination of the large size-classes in a few super-size classes can be misleading and potentially losing information on the real size of the biggest particles.

*(3)Fig 4: The x-axis labels are rather sparse and could be more populated.*

The x-axis is in logarithmic scale; however, ticks were added to have a more populated x-axis.

*(4)Fig 5: Again, only 9 curves are shown for 19 channels listed in the legend, as in Fig. 3. Would it not be better to combine some of these channels for the benefit of clarity?*

As previously replied, the combination of the large size-classes in a few super-size classes can be misleading and potentially losing information on the real size of the biggest particles.

*(5)The arrangement in the table appears somewhat unfortunate to me. The first two rows seem to be unrelated to the remainder of the table and the table deserves a heading to state the unit (nm) for the first column and a symbol with unit for the second column.*

The arrangement of the table is rather customary for a correlation table: the table presents the correlation coefficients between the variables presented in each row (here, ions and particles' number detected in each size range) and those presented in the columns (here, positive and negative ions). The units were added to the table.

*(6)The acknowledgments have distinct font variations disturbing this reader.*

The font variations in the acknowledgements were removed in the revised version of the paper.

---

## Author Comment (AC2) · 16 Apr 2019

Dear Editor,

Please extend our deepest acknowledgements to Anonynous Reviewer #2 for his/her comments on our work.

Please find below in italic the comments received by Anonymous Reviewer #2 after submission to ANGEOD, followed by our replies in normal style.

**Anonymous Referee #2**

*General comments:*

*This study reports the aerosol measurements obtained by the BEXUS18 stratospheric balloon flight. Its distinct feature is being equipped with ion counters. A role of ion chemistry in the stratosphere is still an open question and could be essential for the understanding of the atmospheric impact from the space. This topic is suitable for ANGEO. However, information on the instruments of aerosol and ion measurements is completely lacking in this manuscript, so that it is impossible to evaluate whether their observations are reliable or not. Thus I recommend a rejection of this manuscript.*

We thank the reviewer for his/her comments. The revised version of the manuscript now contains information on the instruments of aerosol and ion measurements, as will be detailed more precisely in the following answers.

*Detailed comments are given below.*

*Specific comments:*

*-Temperature and pressure measurements*

*The authors showed results of temperature and pressure measurement in Fig. 1, but it is found that their results are not so reliable compared to nearby radiosonde observation. Including a radiosonde in their payload does not look difficult, so that I am wondering why they did not do it. In addition, they used temperature and pressure data in their model calculation. Is it really meaningful to use such unreliable data? They need to show how sensitive their model calculation is to temperature and pressure errors.*

In the revised version of the manuscript, we have addressed and provided the results of the sensitivity tests of the model simulations to changes in the T-p profile. In particular, the change in temperature and pressure profile affect only the ion aerosol attachment (approximately 10% change for 20% change in temperature) and the charge particle coagulation coefficient (approximately 6% change for 20% change in temperature and pressure. This does not affect the final model results drastically, as steady state conditions are reached in a couple of hours. Only 1% change in aerosol concentration is observed for an input temperature profile 20% higher/lower in the model input. Because of this, repeating the model calculations with the Kandalashka T-p profile, no significant differences are observed in the final result. To conclude, the T-p profile only changes the rate of the reaction, but not steady-state concentrations. This can be observed from the following Figure, which was added in the Supplementary Material of the revised version of the manuscript.

[Figure]

*-Aerosol measurements*

*Their aerosol instrument, LOAC, has been used in 150 flights, so that its precision, resolution, etc. should be well known. However, they did not give those information at all in the manuscript. Although they mentioned an existence of the thin aerosol layer with a thickness of less than 100m at p.10, l.25, I cannot judge whether this instrument has a vertical resolution high enough to detect such a layer.*

We thank the reviewer for his/her comment. Indeed, this information was missing from the previous version of the manuscript. Instead of applying a smoothing procedure, in the revised version we have integrated the raw measurements over 5 minutes. Thus, we have changed the figures 2 and 3 according to this new procedure. We have added in the text:

"The LOAC vertical resolution is linked to the total concentrations of aerosols, as due to the Poisson counting statistics and the instrument capability to detect the smallest particles. A detailed analysis of the raw measurements has shown that the data must be integrated over 5 minutes to remove the oscillations due to the measurements' uncertainty. Considering the balloon ascent speed, this procedure provides a resolution of about 1 km."

And later:

"To compare directly with the aerosols' measurements, the ions' measurements data are also integrated with a 1-km vertical resolution."

*-Ion measurements*

*They mentioned that the performance of their ion measurement was checked by preflight lab experiments, but it is not shown in the manuscript at all. Thus I cannot judge whether their ion measurements are reliable or not.*

Indications of the performance of the ions' measurements as derived by preflight lab experiments are provided in the revised version of the manuscript, in particular in the form of median absolute deviations separately for positive and negative ions measurements at the 200 mbar pressure level.

*-Average*

*In order to show the aerosol data, they often used arithmetic mean/smoothing. Since aerosol density changes by several orders of magnitude, the arithmetic mean strongly depends on the largest value. Geometrical mean or median filter would be better to represent aerosol distributions*

The aerosols data are now not smoothed (see previous comment).

*-Figs. 3 and 4*

*What is dN/dlog(D) in Fig. 3? A caption of Fig. 4 does not correspond to Fig. 4 about their y-axis.*

dN/dlog(D), or DN/dln(D), represents a commonly used notation in aerosol science to indicate the number concentration of particles in the various size classes (dN) divided by the width of the size classes (dln(D)). The caption of Figures 3 and 4 was slightly modified to indicate more correctly what is shown in the Figure.

*-Eq. (3)*

*Units look different between the terms. Probably some variables are missing.*

The reviewer is right, and Eq. 3 has been corrected in the revised version of the manuscript.

$$\frac{dN_{i,j}}{dt} = \beta_{i-1,j}^{+} N_{i-1,j} n^{+} + \beta_{i+1,j}^{-} N_{i+1,j} n^{-} - \beta_{i,j}^{+} N_{i,j} n^{+} - \beta_{i,j}^{-} N_{i,j} n^{-} + \frac{1}{2} \sum_{l,m=-p}^{l,m=p} \int_{0}^{v} K_{j-v,v}^{l,m} N_{j-v}^{l} N_{v}^{m} dv - $$

$$N_{i,j} \sum_{q=-p}^{p} \int_{0}^{v} K_{j,v}^{i,q} N_{v}^{q} dv \qquad (3)$$

---

## Author Response (AR1)

**Final response for "Measurements of aerosols and charged particles on the BEXUS18 stratospheric ballon"**

Dear Editor,

Thank you very much for providing us the opportunity to submit a revised version of our paper to your journal.

Please find below the comments received by two anonymous reviewers after submission to ANGEOD, followed by our replies and finally our changes in the manuscript.

**1. Comments from Referees**

Anonymous Referee #1

The contribution by Erika Brattich and colleagues reports the measurement and modelling of charged aerosols in the stratosphere. The manuscript is very well written, logically constructed, easy to follow and informative. The manuscript starts with a substantial review of the relevant literature, followed by a thorough theory section that explains the basis for the simulations. Compared to these first two sections, the consecutive section on the experimental results is rather terse and provides little guidance to the reader as to how the individual findings reported in the list of Figures contribute to the key points of the paper. As a result, it would be helpful to expand this section to make the narrative more clear. It is also somewhat surprising that Fig. 5 is not listed in this section, perhaps because it is not considered to be a result of the conducted work. The final sections with the discussion and conclusions emphasise to a large degree the agreement of the findings with previous work. While it is undoubtedly important to put the findings of this study into context, it makes is harder for the reader to appreciate the novelty of the presented work which becomes less clear. It therefore appears to be beneficial for these two sections to distinguish more clearly between known facts and novel findings. Besides this apparent imbalance between the first and second part of the manuscript, I think it is a valuable contribution to the scientific literature as the current knowledge on charged aerosols in the stratosphere and their spatiotemporal variabilities is somewhat limited at present.

Some minor suggestions on how to improve the manuscript are given below.

(1) Fig 2: The concentrations of negative ions appear to be large compared to previous findings. Is there any explanation for this? It is also not clearly explained how the total concentration of aerosols can be smaller than the concentration of negative ions. Is the reader supposed to infer from this that the aerosols

-Aerosol measurements

We thank the reviewer for his/her comment. Indeed, this information was missing from the previous version of the manuscript. Instead of applying a smoothing procedure, in the revised version we have integrated the raw measurements over 5 minutes. Thus, we have changed the figures 2 and 3 according to this new procedure. Explanations for this are now provided in the text.

**-Ion measurements**

Indications of the performance of the ions' measurements as derived by preflight lab experiments are provided in the revised version of the manuscript, in particular in the form of median absolute deviations separately for positive and negative ions measurements at the 200 mbar pressure level.

**-Average**

We thank the reviewer for his/her comment. As previously replied to the comment related to the aerosol measurements, the aerosols data are now not smoothed, thus not averaged.

**-Figs. 3 and 4**

dN/dlog(D), or DN/dln(D), represents a commonly used notation in aerosol science to indicate the number concentration of particles in the various size classes (dN) divided by the width of the size classes (dln(D)).

The caption of Figures 3 and 4 was slightly modified to indicate more correctly what is shown in the Figure.

**-Eq. (3)**

Eq. 3 has been corrected in the revised version of the manuscript.

**3. Author's changes in the manuscript**

We have expanded the Results section inserting also Figure 5 and better presenting our results as shown later in our replies to the reviewers' comments. We have added in the discussion section, page 13, lines 8-11, a summary of our main findings:

"Summarizing, our observations first of all demonstrate the effectiveness of the adopted instrumental setup in measuring vertical profiles of particles' size distributions and particles' typology together with ions. In addition, they can also provide interesting results in terms of the association between cosmic rays and ions, and further to reveal novel features in terms of the charged fraction, from new stratospheric flights with a similar instrumental setup.

(1) We have added in Section 4, page 10, lines 29-32:

"It is important to note first of all that since the lower limit of detection of LOAC is for particles presenting an optical diameter of 200 nm, we can expect to have ions concentrations greater than the

aerosol detected total concentrations. In particular, since the number of ions is greater than the detected aerosol, we can infer that aerosol smaller than 200 nm are the main contributors to negative ions."

(2) We have added better description of the vertical profiles of the different particles' sizes presented in the Figure. The text at page 11, lines 1-8 is now:

"Fig. 3 reports the vertical profiles of aerosol size distribution for each size bin measured by the LOAC instrument. Most of the particles have size below 1  $\mu$ m, as expected in a clean free troposphere and in the stratosphere. Few particles greater than 1  $\mu$ m and smaller than 15  $\mu$ m and just one 50  $\mu$ m particle were detected in the stratosphere. All fine particles (< 1 $\mu$ m) presented the same vertical variation, with a global trend of decreasing concentrations at heights higher than the tropopause. Larger particles, besides presenting lower number concentrations as expected, presented a different vertical profile, with the presence of an abrupt increase in the PBL and then at 10 km less evident in finer particles. However, it is important to note that with these measurements it is difficult to derive information on the PBL, which is out of the focus of this paper."

(3) The x-axis of Figure 4 was populated with more ticks.

(4) Since the combination of the large size-classes in a few super-size classes can be misleading and potentially losing information on the real size of the biggest particles, no change was made.(5) We added the units to the parameters in Table 1.

(6) The font variations in the Acknowledgements were removed in the revised version of the paper.

**-Temperature and pressure measurements**

We have added in the experimental section about numerical simulations (Section 3), page 9, lines 25-26 and page 10, lines 1-6:

"Sensitivity tests to temperature and pressure indicate that the change in temperature and pressure profile affect only the ion aerosol attachment coefficient (approximately 10% change for 20% change in temperature and pressure) and charge particle coagulation coefficient (approximately 6% change for 20% change in temperature and pressure). This does not affect the final model results drastically, as results show that steady state conditions are reached in more or less a couple of hours. Only 1% change in aerosol concentration is observed for an input of 20% higher/lower temperature profile into the model (reported in the Supplementary Material). Overall, no significant differences are observed for 20% change in T-p profile. The T-p profile only changes the rate of the reaction, but not steady state concentrations."

As indicated in the text, a Figure showing the results of sensitivity tests for 20% higher/lower temperature is now provided in the Supplementary Material.

-Aerosol measurements

We have added in the text in Section 2.1, at page 6, lines 19-22:

"The LOAC vertical resolution is linked to the total concentrations of aerosols, as due to the Poisson counting statistics and the instrument capability to detect the smallest particles. A detailed analysis of the raw measurements has shown that the data must be integrated over 5 minutes to remove the oscillations due to the measurements' uncertainty. Considering the balloon ascent speed, this procedure provides a resolution of about 1 km."

And later at page 7, lines 19-20:

"To compare directly with the aerosols' measurements, the ions' measurements data are also integrated with a 1-km vertical resolution."

-Ion measurements

We have added in the text, in Section 2.2, at page 7, lines 7-9:

"Those tests indicate that the average MAD (median absolute deviation) of ions' measurements was equal to 15 ions at 200 mbar for negative ions, and 7 ions for positive ions at the same pressure level."

**-Average**

Modifications to the text are reported in the previous response to the comment on aerosol measurements -Figs. 3 and 4

No change is made but for the captions.

Figure 3: Vertical profiles of particles size distributions for the 19 size classes of the LOAC particle counter as part of the A5-Unibo experiment on the BEXUS18 stratospheric flight. The notation dN/dlog(D) used in the x-axis stands for the number concentration of particles in the various size classes divided by the width of the size classes

Figure 4: Average size distribution of aerosol particles at various height layers during BEXUS18 stratospheric flight. The five curves are obtained averaging the aerosol number densities as a function of the atmospheric layers pointed out by the temperature profile as follows: 1: 0-404 m (black line); 2: 650-1519 m (red line); 3: 1765-10118 m (blue line); 4: 10650-25044 m (pink line); 5: 25289-27191 m (green line). X- and Y-axes are in log-normal scales.

**-Eq. (3)**

The Equation (3) was corrected in the revised version of the manuscript:

 $\frac{dN_{i,j}}{dt} = \beta_{i-1,j}^+ N_{i-1,j} n^+ + \beta_{i+1,j}^- N_{i+1,j} n^- - \beta_{i,j}^+ N_{i,j} n^+ - \beta_{i,j}^- N_{i,j} n^- + \frac{1}{2} \sum_{l,m=-p}^{l,m=p} \int_0^v K_{j-\nu,\nu}^{l,m} N_{j-\nu}^m d\nu - \sum_{l,m=-p}^{l,m=p} \int_0^v K_{j-\nu,\nu}^{l,m} N_{j-\nu,\nu}^m d\nu + \sum_{l=0}^{l,m=p} \int_0^v K_{j-\nu,\nu}^m N_{j-\nu}^m d\nu + \sum_{l=0}^{l,m=p} \int_0^v K_{j-\nu,\nu}^m d\nu + \sum_{l=0}^{l,m=p} \int_0^\infty K_{j-\nu,\nu}^m d\nu + \sum_{l$

 $N_{i,j} \sum_{q=-p}^{p} \int_{0}^{v} K_{j,v}^{i,q} N_{v}^{q} dv$  (3)

**Measurements of Aerosols and Charged Particles On the BEXUS18**

**Stratospheric Balloon**

Erika Brattich1, Encarnación Serrano Castillo2, Fabrizio Giulietti2, Jean-Baptiste Renard3, Sachi N. Tripathi4, Kunal Ghosh4, Gwenael Berthet3, Damien Vignelles3, Laura Tositti5

[revised manuscript text omitted]

(3)

In equation (3), the first two terms on the right-hand side are the probability of interaction between the ions and aerosols of 10 any particular charge and size, and the last term is for growth of that particle due to the charge-particle coagulation process. K is the charge particle coagulation coefficient (probability of collision between two charged particles). v is the aerosol particle volume assuming all particles have a spherical shape,  $N_s$  is the number of aerosol particles for any particular size. Full details are provided in Ghosh et al. (2017).

The model is run for amount of charges of any particular size of particle (q) running from +20 to -20. The ion-aerosol 15 attachment coefficients ( $\beta$ ) are calculated in different ways depending on the relative size of the particles with the ionic mean free path. The calculation depends in particular from the different regimes, i.e. diffusion, free molecular and transition. Hoppel and Frick (1986) developed a method to calculate in all three different regimes. The major requirements for this calculation are the ionic mobility and mean free path, which are calculated using the expressions given by Borucki et al. (1982). The charge coagulation coefficient K was calculated from diffusional force (including vertical diffusion), turbulent

20 shear force, turbulent inertial force along with electrostatic force due to charge on particles (Ghosh et al., 2017).

A polydispersed distribution of aerosols is used in the model and is obtained from the LOAC observation. The LOAC measured data was used for calculation of different input parameters, like ionic mobility, charge particle mobility, charge particle coagulation, ion-aerosol attachment coefficient and ion-ion recombination, which are the global input parameters for the overall model (Global Electrical Circuit Model (GEC), as described in Rawal et al. (2013)). The model also uses temperature and pressure measured during the experiments. Sensitivity tests to temperature and pressure indicate that the change in the input temperature and pressure profile affects only the ion aerosol attachment coefficient (approximately 10%

change for 20% change in temperature and pressure) and charge particle coagulation coefficient (approximately 6% change for 20% change in temperature and pressure). This does not affect the final model results drastically, as results show that steady state conditions are reached in more or less a couple of hours. Only 1% change in aerosol concentration is observed for an input of 20% higher/lower temperature profile into the model (reported in the Supplementary Material). Overall, no

5 significant differences are observed for 20% change in T-p profile. The T-p profile only changes the rate of the reaction, but not steady state concentrations. We added charged particle coagulation model to the Renard et al. (2013) model, as it is close to accurate simulation scenario. The charge balance equations are solved by implicit numerical method to obtain concentrations of positive ions, negative ions, uncharged aerosols and charged aerosols, for the steady state.

**4 Results**

- 10 Fig. 1 shows the comparison between the vertical profiles of relative humidity and temperature measured during the flight with those measured with the radiosonde sounding at Kandalashka (67.15 N, 32.35 E, 25m asl; Russia), 12UTC. The comparison of the temperature profiles shows good agreement in the troposphere with a small inversion layer close to the ground and the starting of the inversion typical of the tropopause located at about 11 km up; The comparison of the temperature profile in the stratosphere presents instead major differences: in fact, while onboard the Bexus flight the 15 temperature remains almost constant in the stratosphere up to an altitude of 26 km and presents a sudden and strong increase around 26-27 km, the temperature profile measured at Kandalashka presents a small decrease until about 25-26 km, typical of profiles of this time of the year in the Arctic region. The reason of such discrepancy in the stratosphere might be due to
- solar heating and perhaps to heat/solar light reflection from other instruments/structures of the gondola. The comparison of the relative humidity profiles presents instead major differences already in the troposphere: in particular, the strong dryness in the Planetary Boundary Layer (PBL) detected onboard the Bexus flight (less than 20%) with a further decrease until the altitude of about 5 km is probably due to the slow response of the relative humidity sensors used onboard. Since standard radiosonde measurements of relative humidity are only reliable in the troposphere above temperatures near -40°C, whereas below these temperatures and in the stratosphere special instrumentation for stratospheric water vapor measurements is

needed (Berthet et al., 2013; Tomikawa et al., 2015), measurements of relative humidity above the tropopause are reported

- 25 only for the Kandalaksha profile, which have to be treated with care nevertheless. Fig. 2 reports the vertical profiles for the cumulative aerosol particle number density obtained by summing up the data from all the size bins collected by the LOAC (in black) together with the negative (blue) and positive (red) ions during the ascent. A sliding smoothing (i.e., each point is simply replaced with the average of m adjacent points) is applied to suppress small scale fluctuations. It is important to note first of all that since the lower limit of detection of LOAC is for particles presenting
- 30 an optical aerodynamic diameter of 200 nm, we can expect to have ions concentrations greater than the aerosol detected total concentrations. In particular, since the number of ions is greater than the detected aerosol, we can infer that aerosol smaller than 200 nm are the main contributors to negative ions.

Fig. 3 reports the vertical profiles of aerosol size distribution for each size bin measured by the LOAC instrument. Most of the particles have size below 1  $\mu$ m, as expected in a clean free troposphere and in the stratosphere. Few particles greater than 1  $\mu$ m and smaller than 15  $\mu$ m and just one 50  $\mu$ m particle were detected in the stratosphere. , which shows that aAll fine particles (< 1 $\mu$ 
[revised manuscript text omitted]

---

## Author Response (AR2)

**Final response for "Measurements of aerosols and charged particles on the BEXUS18 stratospheric ballon"**

Dear Editor,

Thank you very much for providing us the opportunity to submit a revised version of our paper to your journal.

Please find below the comments received by two anonymous reviewers after submission to ANGEOD, followed by our replies and finally our changes in the manuscript.

**1. Comments from Referees**

Anonymous Referee #1

The contribution by Erika Brattich and colleagues reports the measurement and modelling of charged aerosols in the stratosphere. The manuscript is very well written, logically constructed, easy to follow and informative. The manuscript starts with a substantial review of the relevant literature, followed by a thorough theory section that explains the basis for the simulations. Compared to these first two sections, the consecutive section on the experimental results is rather terse and provides little guidance to the reader as to how the individual findings reported in the list of Figures contribute to the key points of the paper. As a result, it would be helpful to expand this section to make the narrative more clear. It is also somewhat surprising that Fig. 5 is not listed in this section, perhaps because it is not considered to be a result of the conducted work. The final sections with the discussion and conclusions emphasise to a large degree the agreement of the findings with previous work. While it is undoubtedly important to put the findings of this study into context, it makes is harder for the reader to appreciate the novelty of the presented work which becomes less clear. It therefore appears to be beneficial for these two sections to distinguish more clearly between known facts and novel findings. Besides this apparent imbalance between the first and second part of the manuscript, I think it is a valuable contribution to the scientific literature as the current knowledge on charged aerosols in the stratosphere and their spatiotemporal variabilities is somewhat limited at present.

Some minor suggestions on how to improve the manuscript are given below.

(1) Fig 2: The concentrations of negative ions appear to be large compared to previous findings. Is there any explanation for this? It is also not clearly explained how the total concentration of aerosols can be smaller than the concentration of negative ions. Is the reader supposed to infer from this that the aerosols <200 nm mainly contribute to the negative ions?

(2) Fig 3: 19 channels are listed in the legend, but only 8 height dependent traces can be distinguished. It is practically impossible to infer any useful information for the PBL.

(3) Fig 4: The x-axis labels are rather sparse and could be more populated.

(4) Fig 5: Again, only 9 curves are shown for 19 channels listed in the legend, as in Fig. 3. Would it not be better to combine some of these channels for the benefit of clarity?

(5) The arrangement in the table appears somewhat unfortunate to me. The first two rows seem to be unrelated to the remainder of the table and the table deserves a heading to state the unit (nm) for the first column and a symbol with unit for the second column.

(6) The acknowledgments have distinct font variations disturbing this reader.

**Anonymous Referee #2**

General comments:

This study reports the aerosol measurements obtained by the BEXUS18 stratospheric balloon flight. Its distinct feature is being equipped with ion counters. A role of ion chemistry in the stratosphere is still an open question and could be essential for the understanding of the atmospheric impact from the space. This topic is suitable for ANGEO. However, information on the instruments of aerosol and ion measurements is completely lacking in this manuscript, so that it is impossible to evaluate whether their observations are reliable or not. Thus I recommend a rejection of this manuscript.

Detailed comments are given below.

Specific comments:

-Temperature and pressure measurements

The authors showed results of temperature and pressure measurement in Fig. 1, but it is found that their results are not so reliable compared to nearby radiosonde observation. Including a radiosonde in their payload does not look difficult, so that I am wondering why they did not do it. In addition, they used temperature and pressure data in their model calculation. Is it really meaningful to use such unreliable data? They need to show how sensitive their model calculation is to temperature and pressure errors.

-Aerosol measurements

Their aerosol instrument, LOAC, has been used in 150 flights, so that its precision, resolution, etc. should be well known. However, they did not give those information at all in the manuscript. Although they mentioned an existence of the thin aerosol layer with a thickness of less than 100m at p.10, l.25, I cannot judge whether this instrument has a vertical resolution high enough to detect such a layer.

-Ion measurements

They mentioned that the performance of their ion measurement was checked by preflight lab experiments, but it is not shown in the manuscript at all. Thus I cannot judge whether their ion measurements are reliable or not.

-Average

In order to show the aerosol data, they often used arithmetic mean/smoothing. Since aerosol density changes by several orders of magnitude, the arithmetic mean strongly depends on the largest value. Geometrical mean or median filter would be better to represent aerosol distributions

-Figs. 3 and 4

What is dN/dlog(D) in Fig. 3? A caption of Fig. 4 does not correspond to Fig. 4 about their y-axis.

-Eq. (3)
Units look different between the terms. Probably some variables are missing.

**2. Author's response**

Anonymous Referee #1
*General comments*

We thank the reviewer for his/her constructive comments. We have included Fig.5 in the narrative of the experimental results section. In addition, the final sections (Sections 4 Results, 5 Discussion, and 6 Conclusions) were revised in accordance to the guidelines provided by the reviewer.

(1) The detection of high concentrations of negative ions is probably due to the fact we added a separate suitable instrument for measuring ions in this flight. Because of the LOAC lower limit of detection of aerosols at 200 nm, we can expect to have ions concentrations greater than the aerosol detected total concentrations due to the presence of aerosols with aerodynamic diameter less than 200 nm. Since the number of ions is greater than the detected aerosol, this indicates that aerosol smaller than 200 nm are the main contributor to negative ions. This comment was added in the revised version of the manuscript.

(2) The text in the revised version of the manuscript was changed to better describe the vertical profiles of the different particles' sizes presented in the Figure. Information on the fact that the information on the PBL, partially commented but out of the scope of the paper, was also added. The combination of the large size-classes in a few super-size classes can be misleading and potentially losing information on the real size of the biggest particles.

(3) The x-axis is in logarithmic scale; however, ticks were added to have a more populated x-axis.

(4) As previously replied, the combination of the large size-classes in a few super-size classes can be misleading and potentially losing information on the real size of the biggest particles.

(5) The arrangement of the table is rather customary for a correlation table: the table presents the correlation coefficients between the variables presented in each row (here, ions and particles' number detected in each size range) and those presented in the columns (here, positive and negative ions). The units were added to the table.

(6) The font variations in the acknowledgements were removed in the revised version of the paper.

General comments:

We thank the reviewer for his/her comments. The revised version of the manuscript now contains information on the instruments of aerosol and ion measurements, as will be detailed more precisely in the following answers.

Specific comments:

-Temperature and pressure measurements

In the revised version of the manuscript, we have addressed and provided the results of the sensitivity tests of the model simulations to changes in the T-p profile. In particular, the change in temperature and pressure profile affect only the ion aerosol attachment (approximately 10% change for 20% change in temperature) and the charge particle coagulation coefficient (approximately 6% change for 20% change in temperature and pressure). This does not affect the final model results drastically, as steady state conditions are reached in a couple of hours. As shown in the next Figure, which presents the results of sensitivity test with 20% difference in the input temperature profile, only 1% change in aerosol concentration is observed for an input temperature profile 20% higher/lower in the model input. Because of this, repeating the model calculations with the Kandalashka T-p profile, no significant differences are observed in the final result. To conclude, the T-p profile only changes the rate of the reaction, but not steady-state concentrations.

[Figure]

-Aerosol measurements

We thank the reviewer for his/her comment. Indeed, this information was missing from the previous version of the manuscript. Instead of applying a smoothing procedure, in the revised version we have integrated the raw measurements over 5 minutes. Thus, we have changed the figures 2 and 3 according to this new procedure. Explanations for this are now provided in the text.

-Ion measurements

Indications of the performance of the ions' measurements as derived by preflight lab experiments are provided in the revised version of the manuscript, in particular in the form of median absolute deviations separately for positive and negative ions measurements at the 200 mbar pressure level.

-Average

We thank the reviewer for his/her comment. As previously replied to the comment related to the aerosol measurements, the aerosols data are now not smoothed, thus not averaged.

-Figs. 3 and 4

dN/dlog(D), or DN/dln(D), represents a commonly used notation in aerosol science to indicate the number concentration of particles in the various size classes (dN) divided by the width of the size classes (dln(D)).

The caption of Figures 3 and 4 was slightly modified to indicate more correctly what is shown in the Figure.

-Eq. (3)
Eq. 3 has been corrected in the revised version of the manuscript.

**3. Author's changes in the manuscript**

We have expanded the Results section inserting also Figure 5 and better presenting our results as shown later in our replies to the reviewers' comments. We have added in the discussion section, page 13, lines 8-11, a summary of our main findings:

"Summarizing, our observations first of all demonstrate the effectiveness of the adopted instrumental setup in measuring vertical profiles of particles' size distributions and particles' typology together with ions. In addition, they can also provide interesting results in terms of the association between cosmic rays and ions, and further to reveal novel features in terms of the charged fraction, from new stratospheric flights with a similar instrumental setup.

(1) We have added in Section 4, page 10, lines 29-32:
"It is important to note first of all that since the lower limit of detection of LOAC is for particles presenting an optical diameter of 200 nm, we can expect to have ions concentrations greater than the

aerosol detected total concentrations. In particular, since the number of ions is greater than the detected aerosol, we can infer that aerosol smaller than 200 nm are the main contributors to negative ions."

(2) We have added better description of the vertical profiles of the different particles' sizes presented in the Figure. The text at page 11, lines 1-8 is now:

"Fig. 3 reports the vertical profiles of aerosol size distribution for each size bin measured by the LOAC instrument. Most of the particles have size below 1 μm, as expected in a clean free troposphere and in the stratosphere. Few particles greater than 1 μm and smaller than 15 μm and just one 50 μm particle were detected in the stratosphere. All fine particles (< 1μm) presented the same vertical variation, with a global trend of decreasing concentrations at heights higher than the tropopause. Larger particles, besides presenting lower number concentrations as expected, presented a different vertical profile, with the presence of an abrupt increase in the PBL and then at 10 km less evident in finer particles. However, it is important to note that with these measurements it is difficult to derive information on the PBL, which is out of the focus of this paper."

(3) The x-axis of Figure 4 was populated with more ticks.

(4) Since the combination of the large size-classes in a few super-size classes can be misleading and potentially losing information on the real size of the biggest particles, no change was made.
(5) We added the units to the parameters in Table 1.

(6) The font variations in the Acknowledgements were removed in the revised version of the paper.

-Temperature and pressure measurements
We have added in the experimental section about numerical simulations (Section 3), page 9, lines 25-26 and page 10, lines 1-6:

"Sensitivity tests to temperature and pressure indicate that the change in temperature and pressure profile affect only the ion aerosol attachment coefficient (approximately 10% change for 20% change in temperature and pressure) and charge particle coagulation coefficient (approximately 6% change for 20% change in temperature and pressure). This does not affect the final model results drastically, as results show that steady state conditions are reached in more or less a couple of hours. Only 1% change in aerosol concentration is observed for an input of 20% higher/lower temperature profile into the model (reported in the Supplementary Material). Overall, no significant differences are observed for 20% change in T-p profile. The T-p profile only changes the rate of the reaction, but not steady state concentrations."
As indicated in the text, a Figure showing the results of sensitivity tests for 20% higher/lower temperature is now provided in the Supplementary Material.

-Aerosol measurements

We have added in the text in Section 2.1, at page 6, lines 19-22:

"The LOAC vertical resolution is linked to the total concentrations of aerosols, as due to the Poisson counting statistics and the instrument capability to detect the smallest particles. A detailed analysis of the raw measurements has shown that the data must be integrated over 5 minutes to remove the oscillations due to the measurements' uncertainty. Considering the balloon ascent speed, this procedure provides a resolution of about 1 km."

And later at page 7, lines 19-20:

"To compare directly with the aerosols' measurements, the ions' measurements data are also integrated with a 1-km vertical resolution."

-Ion measurements

We have added in the text, in Section 2.2, at page 7, lines 7-9:

"Those tests indicate that the average MAD (median absolute deviation) of ions' measurements was equal to 15 ions at 200 mbar for negative ions, and 7 ions for positive ions at the same pressure level."

-Average

Modifications to the text are reported in the previous response to the comment on aerosol measurements

-Figs. 3 and 4

No change is made but for the captions.

Figure 3: Vertical profiles of particles size distributions for the 19 size classes of the LOAC particle counter as part of the A5-Unibo experiment on the BEXUS18 stratospheric flight. The notation dN/dlog(D) used in the x-axis stands for the number concentration of particles in the various size classes divided by the width of the size classes

Figure 4: Average size distribution of aerosol particles at various height layers during BEXUS18 stratospheric flight. The five curves are obtained averaging the aerosol number densities as a function of the atmospheric layers pointed out by the temperature profile as follows: 1: 0-404 m (black line); 2: 650-1519 m (red line); 3: 1765-10118 m (blue line); 4: 10650-25044 m (pink line); 5: 25289-27191 m (green line). X- and Y-axes are in log-normal scales.

-Eq. (3)

The Equation (3) was corrected in the revised version of the manuscript:

$$\frac{dN_{i,j}}{dt} = \beta_{i-1,j}^{+} N_{i-1,j} n^{+} + \beta_{i+1,j}^{-} N_{i+1,j} n^{-} - \beta_{i,j}^{+} N_{i,j} n^{+} - \beta_{i,j}^{-} N_{i,j} n^{-} + \frac{1}{2} \sum_{l,m=-p}^{l,m=p} \int_{0}^{v} K_{j-v,v}^{l,m} N_{j-v}^{l} N_{v}^{m} dv -$$

$$N_{i,j} \sum_{q=-p}^{p} \int_{0}^{v} K_{j,v}^{i,q} N_{v}^{q} dv \qquad (3)$$